# The Effects of Leisure Activities on Self-Efficacy and Social Adjustment: A Study of Immigrants in South Korea

**DOI:** 10.3390/ijerph18168311

**Published:** 2021-08-05

**Authors:** Chul-Ho Bum, Joon-Hee Lee, Chulhwan Choi

**Affiliations:** 1Department of Golf Industry, College of Physical Education, Kyung Hee University, Seocheon-dong 1, Giheung-gu, Yongin-si 17104, Korea; bch2180@khu.ac.kr; 2Department of Coaching, College of Physical Education, Kyung Hee University, Seocheon-dong 1, Giheung-gu, Yongin-si 17104, Korea; borracho@khu.ac.kr; 3Department of Physical Education, Gachon University, 1342, Seongnam-daero, Sujeong-gu, Seongnam-si 13120, Korea

**Keywords:** immigrant, leisure activity type, self-efficacy, social adjustment

## Abstract

This study classified leisure activity types into active, passive, and social leisure activities based on theory, and focused on determining the type that has a significant influence on the self-efficacy and social adjustment of immigrants staying in South Korea. The results of multivariate analysis of variance (MANOVA), including post-hoc analysis using SPSS 23.0, were as follows: in principle, immigrants who participate in active or social leisure activities perceive their self-efficacy and social adjustment to be high. Differing slightly from this, the passive leisure activity type, which includes activities such as reading alone, listening to music, and surfing the web, may relieve their stress or provide them with psychological stability, but it was not found to be helpful in their adjustment to the new culture. The significance of this study lies in the finding that leisure activities help immigrants with social adjustment, in addition to physical and psychological aids that are already well known. We hope that the findings of the present study can be used as basic data for helping immigrants with smooth social adjustment and increasing their quality of life.

## 1. Introduction

In the era of globalization, the number of global migrants is gradually increasing, and Asian countries are experiencing a transformation from previously homogeneous populations to multicultural societies [1]. This means that unlike Western countries, many changes are occurring in many areas, such as ethnic identity in Asian countries, where mono-ethnicity has been the norm [2]. In relation to this, according to the Ministry of Justice of the Republic of Korea [3], the number of immigrants crossed 1 million in 2007 and exceeded 2 million in the following 9 years. The number of immigrants is expected to exceed 3 million in 2021 (5.82% of the total population) based on the current trend [4]. The major reasons for this consistent increase in the number of immigrants were employment, marriage immigration, and student migration [4]. A considerable number of immigrants may face difficulties in adapting to a new society with different lifestyles, cultures, environments, etc. In addition to these problems, a recent survey found that seven out of 10 immigrants reported racism in Korea. The reasons for discrimination, including “not Korean” and “not fluent in Korean”, were found to be more frequent than race, ethnicity, and skin colors [5]. These difficulties have been reported to cause stress or psychological withdrawal [6]. If these negative psychological conditions persist and are not improved, it will inevitably decrease their self-efficacy and create difficulties in social adaptation.

Previous research has discussed the need to bridge gaps in culture and customs, and increase the linguistic competence of immigrants to help them adjust to the new society. However, socialization and frequent contact with community members can be considered as the most important factors in this adjustment [7,8]. Kim and Lee [9] contended that the participation of immigrants in leisure activities helps form a strong network with various locals who share the same interests, affirm their existence, and increase their quality of life. In particular, leisure sports activities are advantageous, as they can provide a space for interaction and communication with other local residents from different linguistic or cultural backgrounds [10,11,12].

However, immigrants are not only discriminated in the “labor market” but also face various restrictions on participating in “leisure activities,” which are unique areas for individuals [2]. Therefore, the concept of leisure needs to be analyzed in greater detail since it includes not only physical activities but also various recreational activities of individuals. Floyd [13] defined leisure activities as everything that individuals voluntarily participate in, to pursue fun or enjoyment outside economic activity. Past research has primarily focused on the analysis of active leisure through physical activities, and all their findings were positive for the participants [14,15]. Studies on passive leisure in which individuals pursue enjoyment (e.g., watching TV) while spending comfortable private time began to emerge, and some defined it as sedentary [16] or relaxed leisure [17]. According to Csikszentmihalyi and Hunter [18], the possibility of negative influence of passive leisure on healthy daily living has been raised, but passive leisure has nonetheless been considered a leisure activity. The last element is social leisure, which is enjoyed by spending time with friends or family [19]. Studies have analyzed the social relationships of adolescents who value peer relationships [20] or the elderly who enjoy life after retirement [21]. It is important for the present study to investigate the role of social leisure activities as an important tool for immigrants who need social relationships to adjust to living in a foreign country.

### 1.1. Self-Efficacy

Self-efficacy refers to the belief or faith in one’s own ability to successfully perform a certain action in a specific situation [22]. In this respect, people with high self-efficacy often have motivations that play an important role in achieving their goals [23,24]. Albert Bandura [22], in his model, explained four sources of developing self-efficacy beliefs. First, performance accomplishment experiences can increase self-efficacy. These experiences enable an individual to gain self-confidence regarding a task when they see others perform it successfully. Conversely, self-efficacy decreases when an individual observes others’ failure to perform. Indeed, there is a significantly positive relationship between performance accomplishment experiences and self-efficacy in the field of sports [25,26].

This model clearly states the effect of vicarious experience even though its effect may be relatively weaker than one’s own experience [27,28,29]. When faced with difficult challenges or unsure of performing a particular task, the vicarious experience allows individuals to observe other people succeed and strengthen their self-efficacy beliefs in this process [30]. In addition, verbal persuasion or positive feedback from people around the individual is discussed as a method of assuring success [31,32]. Verbal persuasion means encouraging people by speaking a positive language or using compliments [33].

Additionally, emotional and physiological states can change the level of self-efficacy. How people interpret their emotive states can affect their sense of self-effectiveness [34]. For example, feeling positive emotions, such as fun, satisfaction, fulfillment, and joy, while participating in sports activities could create positive psychological effects [21,35]. In contrast, negative emotions, such as anxiety, tension, worry, and depression, degrade self-effectiveness. People with high self-efficacy do not easily give up and tend to overcome difficult situations [36,37], while people with low self-efficacy perceive problems as being more difficult than their actual level of difficulty. Therefore, they suffer high levels of stress [38]. Interestingly, people with high self-efficacy were found to adjust themselves relatively well to the group to which they belonged [39,40].

### 1.2. Social Adjustment

Adjustment is defined as the process of changing oneself to fit into a given environment [41]. In such a context, the social adjustment of immigrants refers to the level of understanding of the culture or customs of the new country in which they settle and live without psychological discomfort [42,43]. Nonetheless, a considerable number of immigrants experience a significant amount of stress in the process of moving into a new place where culture, customs, and language are unfamiliar and different from their original country of residence [44]. Migrant workers experience a sense of loss due to perceived discrimination or identity confusion, which can lead to mental health problems such as depression or anxiety [45,46,47]. In the present study, social adjustment was classified into four elements: Communication, interpersonal, cultural, and emotional adjustments. Previous studies have reported that frequent contact with locals is an important element for the smooth social adjustment of immigrants [7,8], and to that end, participation in leisure activities is helpful, allowing immigrants to form networks with locals and learn about the new culture and its customs [9]. Therefore, this study analyzed the differences between the self-efficacy and social adaptation of immigrants in Korea by classifying the type of leisure activities into active, passive, and social activities based on the previous literature. This study provides in-depth information necessary to enhance the quality of individuals’ lives by improving the psychological well-being and social adaptation of immigrants who are struggling in other countries.

## 2. Materials and Methods

### 2.1. Participants

The present study aimed to analyze differences in the self-efficacy and social adjustment of immigrants staying in South Korea for an extended period, according to the types of leisure activities. The survey respondents were limited to adult immigrants aged 20 years or older, residing in South Korea for at least 6 months. The reason for defining a stay of 6 months was that immigrants who stayed for a short time for purposes such as tourism, did not fit the purpose of this study. Accordingly, a “long-stay” was defined as staying for 6 months or longer, which was the criterion for immigrants to purchase a health insurance policy. Additionally, the nationality and age of the participants were confirmed by their identifications. Participation in the present study was voluntary, and the participants were made sufficiently aware of their rights at the outset, including their right to stop participating at any time, even if they had neared the end of the questionnaire.

Data were collected in two places in downtown Seoul for a total of 2 months, from March to April 2019. G*Power (effect size = 0.0625, *p* = 0.05, power = 0.8) was used to calculate a satisfactory sample size (greater than *n* = 153) for this study [48]. A total of 366 questionnaires were collected from the 600 questionnaires distributed (response rate: 66.0%), and 323 questionnaires were used in the final analysis, after excluding incomplete questionnaires. The inclusion of survey participants, in the present study, was first determined by asking questions about the duration of the stay. Additionally, they answered demographic questions and the types of leisure activities they preferred and enjoyed. Based on the results, the participants were grouped into three groups: (a) Group 1: Active leisure group (*n* = 104, 32.2%), (b) Group 2: Passive leisure group (*n* = 91, 28.2%), and (c) Group 3: Social leisure group (*n* = 128, 39.6%). Detailed information on the descriptive statistics is provided in Table 1.

### 2.2. Instruments

To investigate the self-efficacy of the participants in this study based on the leisure activity type, an instrument (three factors with 12 items) by Shin [49] examining self-efficacy of the elderly after retirement showed satisfactory psychometric properties in a previous study (social efficacy (four items), *α* = 0.849; personal achievement (four items), *α* = 0.717; self-acceptance (four items), *α* = 0.812). However, the instrument was modified and applied in accordance with the purpose of this study (i.e., deleting three items). As a result, this study utilized an instrument consisting of three factors with nine items.

Next, to examine the social adjustment of immigrants, a measurement tool by Yang [50] investigated the effects of spouses’ values on the social adjustment of immigrants. It also revealed adequate psychometric properties in the previous study as follows: Interpersonal relationship adjustment (four items): *α* = 0.812; communication adjustment (five items): *α* = 849; emotional adjustment (three items): *α* = 583; and cultural adjustment (five items): *α* = 849. Depending on the purpose of this study, after excluding two items, four factors with 15 items were finally applied in this study. Additionally, questionnaires formed by negative sentences on communication and emotional adjustment factors were converted using reverse coding for consistent results. All of the items utilized five-point Likert-type scales ranging from “strongly disagree” (1 point) to “strongly agree” (5 points).

### 2.3. Data Analysis

Data analysis was conducted using the SPSS 23.0. First, the analysis provided descriptive statistics, including the socio-demographic information of the survey respondents. Next, to ensure the validity of the data collected in this study, two exploratory factor analyses (EFAs) for each dependent variable (i.e., self-efficacy and social adjustment) were conducted. To ensure the reliability of the data utilized in this study, Cronbach’s alpha coefficients were applied. Finally, a multivariate analysis of variance (MANOVA) including the post-hoc analysis was conducted to compare the differences in dependent variables (i.e., social efficacy, personal achievement, self-acceptance, interpersonal relationship adjustment, communication adjustment, emotional adjustment, and cultural adjustment) based on the three groups (leisure activity types).

## 3. Results

### 3.1. Validity and Reliability

The two EFAs with a principal component analysis (PCA) were implemented on each dependent variable: (a) Self-efficacy (social efficacy (three items), personal achievement (three items), and self-acceptance (three items)), and (b) social adjustment (interpersonal relationship adjustment (four items), communication adjustment (five items), emotional adjustment (three items), and cultural adjustment (three items)).

Regarding the factor structure of the self-efficacy factor, the Kaiser–Meyer–Olkin (KMO) measure revealed the sample adequacy (0.649), which was between 0.60 and 0.70 [51]. Bartlett’s test of sphericity was statistically significant (*χ^2^* = 968.254, *df* = 36, *p* < 0.01). The retained three factors, accounting for 70.441% of the total variance, had eigenvalues greater than one and factor structure coefficient greater than 0.40. In addition, the factors indicated acceptable Cronbach’s alpha coefficients for internal consistency greater than 0.70, as follows: Social efficacy (*α* = 0.821), personal achievement (*α* = 0.753), and self-acceptance (*α* = 0.767) [52] (Table 2).

Next, regarding the factor structure of the social adjustment factor, the Kaiser–Meyer–Olkin (KMO) measure showed a sample adequacy (0.746) greater than 0.70 [51]. Bartlett’s test of sphericity was also statistically significant (*χ^2^* = 1990.475, *df* = 105, *p* < 0.01). The retained four factors, accounting for 67.059% of the total variance, also revealed eigenvalues greater than one and factor structure coefficients greater than 0.40. Finally, the factors ensured acceptable Cronbach’s alpha coefficients for internal consistency greater than 0.70, as follows: Interpersonal relationship adjustment (*α* = 0.898), communication adjustment (*α* = 0.826), emotional adjustment (*α* = 0.738), and cultural adjustment (*α* = 0.703) [52] (Table 3).

### 3.2. Multivariate Analysis of Variance (MANOVA)

As shown in Table 4, the MANOVA found statistically significant differences in the seven dependent variables depending on the type of leisure activities (Wilks’ lambda = 0.735, *F* (14, 628) = 7.469, *p* = 0.00, partial *η*^2^ = 0.143). More specifically, the univariate tests showed statistically significant effects for social efficacy and self-acceptance on the self-efficacy factor, and communication adjustment, interpersonal relationship adjustment, and emotional adjustment on the social adjustment factor. No significant differences were found in personal achievement or cultural adjustment items from the analyses.

To verify where statistically significant differences existed among more than three groups, additional post-hoc analyses were performed. Regarding social efficacy, self-acceptance, interpersonal relationship adjustment, and emotional adjustment factors, the active and social leisure participant groups (Groups 1 and 3) had relatively higher mean scores than the passive leisure participant group (Group 2). Regarding the communication adjustment factor, the social leisure participant group (Group 3) showed statistically significant higher mean scores than the other groups (Groups 1 and 2). More detailed information on the mean scores of each group on the dependent variables is provided in Table 5 and Table 6.

## 4. Discussion

The present study focused on analyzing the differences in self-efficacy and social adjustment of immigrants according to the types of leisure activities (active, passive, and social). First, the results related to self-efficacy showed that immigrants who participated in active and social leisure activities perceived their social self-efficacy and self-acceptance to be high. In general, immigrants experience stress or psychological withdrawal when they encounter a new culture, lifestyle or language [6]. In such an environment, participation in leisure activities resolves physical or psychological fatigue and provides opportunities for self-realization [53,54]. Accordingly, a study conducted by Iso-Ahola [55] reported that voluntary participation in leisure activities improves overall satisfaction with life by increasing self-efficacy. It is noteworthy that the findings that physical activity protects positive mental health are supported not only in Eastern cultures [56,57] but also in Western cultures [58]. Considering that the present study found similar results regarding the participation of immigrants in leisure activities, physical activities should be encouraged not only for physical health but also for psychological aspects.

Furthermore, the social leisure group also showed a high level of social efficacy, indicating the importance of establishing social relationships. This reveals that while participating in leisure activities, an individual forms networks with other people, thus reducing the feelings of social isolation. There are several previous studies that reported the negative effects of social isolation on physical and psychological conditions [59,60,61,62]. It inevitably has a significant influence on lowering the quality of life [63,64,65,66]. There exists a consensus in the literature that social isolation is one of the most difficult problems faced by immigrants [67]. In the case of immigrants, the importance of social leisure for the formation of social relationships appears to be even greater.

Similar results were found in the analysis of the social adjustment of immigrants who participated in leisure activities. In particular, those who participated in active and social leisure activities showed relatively high results for interpersonal relationships and emotional adjustment factors, while those who participated in social leisure activities showed exceptionally higher results than the other two types of leisure for the communication factor. For immigrants living in areas where culture, customs, and language are different, contact and socialization with locals are important for smooth social adjustment [7,8]. That is, leisure activities with locals provide a place for interaction and communication, and help them with cultural or emotional adjustment [10,11,12]. The findings of these previous studies support those of the present study. However, the results of passive leisure activities were relatively low in all the areas. Passive leisure activities are also called sedentary behaviors, which include playing computer games, surfing the web, and listening to music alone at home [16]. It is significant that these leisure activities, which can be done alone at home, do not provide much support for social adjustment.

In general, contact theory is a theory for improving intergroup relations among minorities, such as the disabled people in society [68]. The immigrants examined in the present study also faced difficulties as a minority, while they were adjusting to life as members of society in a different culture. The concept of contact theory is that bias is reduced and relationships are improved through intergroup contacts, and it is frequently cited when relationships between majority and minority groups in society are discussed [69]. We may conclude that they have a more stable adjustment than immigrants who enjoy leisure activities and have many social contacts, which is consistent with the concept of contact theory. Accordingly, they can have more contact opportunities as members of society, not only through physical leisure activities but also through social leisure activities, which can positively affect the self-efficacy and social adjustment of immigrants. Specific measures at the government or community level will be of great help to move beyond individual efforts to address the increasing number of immigrants.

## 5. Conclusions and Limitations

Recently, the number of immigrants has been increasing every day, due to the influx of international marriages, immigrants, and foreign workers, even in South Korea. However, a considerable number of immigrants experience stress due to the unfamiliar culture, customs or lifestyles as they move to a new location, causing psychological withdrawal [6]. In this study, it was found that sharing leisure activities with locals who have common interests in a social environment can enhance self-efficacy and social adjustment. In particular, among the many types of leisure activities, active or social leisure activities were found to provide significant help. Meanwhile, passive leisure activities such as watching TV, reading or surfing the web provided psychological well-being, but were insignificant in aiding self-efficacy and social adjustment.

The suggestions based on the present study are as follows: First, the present study was conducted on immigrants in South Korea, but did not consider the possible effect of changes in the duration of their stay on the results. Future studies should consider this limitation. Furthermore, the present study measured the self-efficacy or social adjustment of immigrants using a questionnaire. Future studies should consider observations and interviews conducted in parallel with the survey. Lastly, the data in this study were not normally distributed, violating the normality assumption for the multivariate test. According to Field [70], it might be a product of non-normal data. Furthermore, Stevens [71] stated that the multivariate analyses are robust to violations of the assumption given relatively equal group sizes, as was the case in the current study.

## Figures and Tables

**Table 1 ijerph-18-08311-t001:** Descriptive statistics by groups.

		Group 1	Group 2	Group 3
		Active	Passive	Social
Gender	Male	70 (67.3%)	44 (48.4%)	77 (60.2%)
	Female	34 (32.7%)	47 (51.6%)	51 (39.8%)
Age	20s	25 (24.0%)	30 (33.0%)	44 (34.4%)
	30s	26 (25.0%)	22 (24.2%)	26 (20.3%)
	40s	30 (28.8%)	14 (15.4%)	27 (21.1%)
	50s	16 (15.4%)	17 (18.7%)	23 (18.0%)
	Over 60s	7 (6.7%)	8 (8.8%)	8 (6.3%)
Nationality	China	47 (45.2%)	40 (44.0%)	61 (47.7%)
	Vietnam	16 (15.4%)	17 (18.7%)	23 (18.0%)
	Philippine	14 (13.5%)	14 (15.4%)	12 (9.4%)
	United States	13 (12.5%)	12 (13.2%)	18 (14.1%)
	Japan	7 (6.7%)	6 (6.6%)	10 (7.8%)
	Etc.	7 (6.7%)	2 (2.2%)	4 (3.1%)
Length of stay	6 months–1 year	46 (44.2%)	45 (49.5%)	72 (56.3%)
	1–5 years	20 (19.2%)	20 (22.0%)	33 (25.8%)
	Over 5 years	38 (36.5%)	26 (28.6%)	23 (18.0%)
Active leisure activity	Walking	31 (29.8%)	-	-
	Gym workout	22 (21.2%)	-	-
	Hiking	25 (24.0%)	-	-
	Cycling	11 (10.6%)	-	-
	Swimming	8 (7.7%)	-	-
	Badminton	3 (2.9%)	-	-
	Table Tennis	4 (3.8%)	-	-
Passive leisure activity	Web Surfing	-	49 (53.8%)	-
	TV watching	-	30 (33.0%)	-
	Listening to music	-	6 (6.6%)	-
	Reading books	-	6 (6.6%)	-
Social leisure activity	Meeting people	-	-	128 (100%)
Totals		104 (100%)	91 (100%)	128 (100%)

Source: Own study.

**Table 2 ijerph-18-08311-t002:** Factor structure matrix for self-efficacy.

Items	1	2	3
I have good relationships with my acquaintances	**0.907**	−0.003	0.037
I have someone with whom I can have a heart-to-heart talk	**0.881**	−0.037	−0.036
I am needed by people around me	**0.789**	0.048	0.019
I am a valuable person	−0.012	**0.875**	0.033
I am a successful person	−0.032	**0.868**	0.004
I have a lot to be proud of	0.049	**0.742**	−0.007
I have hope in life	0.095	0.022	**0.859**
I have work to do every day	0.011	−0.015	**0.850**
I always learn something	−0.073	0.020	**0.748**
Eigenvalues	2.253	2.100	1.987
Variance (%)	25.030	23.338	22.074
Cronbach’s alpha	0.821	0.753	0.767

Note. 1 = social efficacy, 2 = personal achievement, 3 = self-acceptance.

**Table 3 ijerph-18-08311-t003:** Factor structure matrix for social adjustment.

Items	1	2	3	4
My close acquaintances and I help each other	**0.916**	0.023	0.071	0.048
I have local friends to open my mind to things	**0.877**	0.074	0.022	0.021
I make local friends easily	**0.864**	0.033	0.080	0.031
I am comfortable with the local people	**0.835**	−0.018	−0.057	0.015
I have difficulties due to the unfamiliar language	0.002	**0.867**	0.126	0.067
Using respectful words is difficult	−0.074	**0.791**	−0.100	0.050
Jokes are difficult to understand	0.117	**0.735**	0.097	0.069
It is difficult to think in a foreign language	0.046	**0.729**	0.070	−0.011
Expressing myself is difficult	0.018	**0.710**	−0.073	−0.013
I shun contact with people	0.064	0.027	**0.876**	0.010
People are biased against me	0.043	0.066	**0.855**	0.059
My social status is low as an immigrant	−0.011	−0.007	**0.681**	−0.173
I understand the local political culture	0.047	0.087	−0.088	**0.838**
I understand the local food culture	0.026	−0.039	−0.120	**0.822**
I understand the local popular culture	0.020	0.067	0.079	**0.722**
Eigenvalues	3.328	2.845	2.162	1.724
Variance (%)	22.189	18.964	14.416	11.490
Cronbach’s alpha	0.898	0.826	0.738	0.703

Note. 1 = interpersonal relationship adjustment, 2 = communication adjustment, 3 = emotional adjustment, 4 = cultural adjustment.

**Table 4 ijerph-18-08311-t004:** Results of the MANOVA.

Variables	Sub-Factors	*df*	*F*	*p*	*η^2^*
Self-efficacy	Social efficacy	2	12.020	0.000 *	0.070
	Self-acceptance	2	6.288	0.002 *	0.038
	Personal achievement	2	0.231	0.794	0.001
Social adjustment	Communication adjustment	2	8.649	0.000 *	0.051
	Interpersonal relationship adjustment	2	15.336	0.000 *	0.087
	Cultural adjustment	2	0.246	0.782	0.002
	Emotional adjustment	2	15.636	0.000 *	0.089

* *p* < 0.05.

**Table 5 ijerph-18-08311-t005:** Mean scores of dependent variables based on groups.

	Self-Efficacy			Social Adjustment			
	1	2	3	4	5	6	7
Group 1	**3.249**	**3.176**	2.840	2.944	**3.060**	2.930	**3.061**
Group 2	2.725	2.773	2.854	2.842	2.420	2.993	2.359
Group 3	**3.284**	**3.190**	2.920	**3.273**	**3.111**	3.005	**2.883**

Note. Group 1 = active leisure participants, Group 2 = passive leisure participants, Group 3 = social leisure participants; 1 = social efficacy, 2 = personal achievement, 3 = self-acceptance, 4 = interpersonal relationship adjustment, 5 = communication adjustment, 6 = emotional adjustment, 7 = cultural adjustment; and statistically significant higher mean scores among groups are indicated in bold.

**Table 6 ijerph-18-08311-t006:** Results of post-hoc analysis.

		Self-Efficacy			Social Adjustment			
		1	2	3	4	5	6	7
G1	G2	0.000 *	0.008 *	0.994	0.656	0.000 *	0.864	0.000 *
	G3	0.947	0.993	0.804	0.007 *	0.917	0.780	0.297
G2	G1	0.000 *	0.008 *	0.994	0.656	0.000 *	0.864	0.000 *
	G3	0.000 *	0.004 *	0.871	0.000 *	0.000 *	0.994	0.000 *
G3	G1	0.947	0.993	0.804	0.007 *	0.917	0.780	0.297
	G2	0.000 *	0.004 *	0.871	0.000 *	0.000 *	0.994	0.000 *

Note. * *p* < 0.05, G1 = active leisure participants, G2 = passive leisure participants, G3 = social leisure participants; 1 = social efficacy, 2 = personal achievement, 3 = self-acceptance, 4 = interpersonal relationship adjustment, 5 = communication adjustment, 6 = emotional adjustment, and 7 = cultural adjustment.

## Data Availability

Restrictions apply to the availability of these data.

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
