# Peer review of "The Effects of Leisure Activities on Self-Efficacy and Social Adjustment: A Study of Immigrants in South Korea"

_ijerph, 2021, doi:10.3390/ijerph18168311_

Round 1
Reviewer 1 Report
It is noted that two modified instruments (para 2.2) have been used in the present study. Shin's one is for the elderly after retirement and Yang's one is for spouses. More explanation will be required for why these instruments are valid, reliable and appropriate as the participants of this study are male and female and most of them are under 60s.
In addition, this study utilized Bandura's self-efficacy theory (1.1). More in-depth elaboration of his four sources of efficacy beliefs will be beneficial.
Please delete the word "and" in line 276.
Author Response
Reviewer 1.
- It is noted that two modified instruments (para 2.2) have been used in the present study. Shin's one is for the elderly after retirement and Yang's one is for spouses. More explanation will be required for why these instruments are valid, reliable and appropriate as the participants of this study are male and female and most of them are under 60s. We absolutely understand your concern. However, the measurements applied in this study from previous research were not intended for specific research samples (i.e., elderly or female). Additionally, a total of three professors in physical education, including one professor in sports psychology, were confirmed through the content validation process that there was no problem measuring self-efficacy and social adjustment of survey respondents who participated in the study.
- In addition, this study utilized Bandura's self-efficacy theory (1.1). More in-depth elaboration of his four sources of efficacy beliefs will be beneficial. Based on your comment, the section of Introduction has been revised and supplemented.
- Please delete the word "and" in line 276. Based on your comment, we have deleted the word.

Reviewer 2 Report
I want to express my gratitude to the authors for conducting such an intriguing study. The purpose of this study was to examine the role of social leisure activities as a critical tool for foreigners seeking to adjust to life in a foreign country. Additionally, numerous issues with the research were observed. For instance, it appears as though the text's explanations are unbalanced.
- I suggest sending the manuscript to an English editor for proofreading.
- Title: Suggest that foreign residents be used instead of resident foreigners.
- The introduction should include a stronger justification for the study's purpose.
- The review of the literature was insufficient; it lacked a conceptual framework.
- Kindly include a sample size calculation. Please provide effect size and also reference as justification.
- Is the questionnaire in English or Korean? Please provide the method of translation to Korean language (if in Korean language) and provide the validity and reliability information on the questionnaire used in this study.
- Please verify the version of SPSS you are using. Version 23.0 is no longer supported.
- Why did the authors perform EFA before performing MANOVA? EFA should use a different sample size than MANOVA. Separate samples should be used for EFA and MANOVA.
- Please include information about verifying the multivariate normality assumption in statistical analysis, including the various fit indices used and their recommended values. Kindly provide references to substantiate your claim.
- Please include ethical approval and procedure to approach participants.
- Please create a flow chart for the study using Consort.
- Please discuss further on previous research and compare it to the study's findings in the Discussion section.
- I felt that the study's justification, conceptual framework, and information data analysis were insufficient. By omitting these details, the reader is left with more questions than answers.
Best wishes.
Many thanks.
Author Response
Reviewer 2.
- I suggest sending the manuscript to an English editor for proofreading. Based on your comment, we have worked with English proofreaders.
- Title: Suggest that foreign residents be used instead of resident foreigners. Based on your comment, we have revised the title.
- The introduction should include a stronger justification for the study's purpose. Based on your comment, the section of Introduction has been revised and supplemented.
- The review of the literature was insufficient; it lacked a conceptual framework. Based on your comment, the section of Introduction (the latter part) has been revised and supplemented.
- Kindly include a sample size calculation. Please provide effect size and also reference as justification. Based on G*power (effect size = .0625; p = .05; power = .8), we have verified a satisfactory sample size (greater than n = 153). As a result, 323 questionnaires were used in the final analysis. We have added the detailed information in the section of 1. Participants.
- Is the questionnaire in English or Korean? Please provide the method of translation to Korean language (if in Korean language) and provide the validity and reliability information on the questionnaire used in this study. The questionnaire was written in English and Korean, and respondents chose their preferred language. As mentioned in this study, the survey respondents were limited to adult foreigners residing in South Korea for at least six months. The reason for defining a stay of six months was that foreigners who stayed for a short time for purposes did not fit the purpose of this study. As a result, during the data collection, there was no problem with language barriers.
- Please verify the version of SPSS you are using. Version 23.0 is no longer supported. Based on your comment, we have analyzed collected data via SPSS 25.0.
- Why did the authors perform EFA? EFA should use a different sample size than MANOVA. Separate samples should be used for EFA and MANOVA. In this study, the EFA was conducted to secure validity of the instruments before performing MANOVA. According to Yu and Richard (2015), EFA is a statistical evidence of validity. Also, Wetzel (2011) stated that EFA is performed for revised instrument.
- Please include information about verifying the multivariate normality assumption in statistical analysis, including the various fit indices used and their recommended values. Kindly provide references to substantiate your claim. The data in this study was not normally distributed, resulting in the violation of the normality assumption for the multivariate test. However data in this study was not normally distributed, resulting in the violation of the normality assumption for multivariate test. However, according to Stevens (2002) and Field (2009), this may have been a product of non-normal data. Stevens (2002) stated that multivariate analyses are robust to violations of the assumption given relatively equal group sizes, as was the case in the current study. We have added detailed information in the section of Conclusion and Limitation.
- Please include ethical approval and procedure to approach participants. This study not collecting sensitive information of respondents was conducted without IRB's approval as a survey research in the field of social science. In such cases, IRB review is not mandatory conventionally in Korea. However, as mentioned in the manuscript, the survey respondents were fully explained in advance about the purpose of the study and their rights. Participants volunteered to participate and could be discontinued at any time during the survey. I have attached a file used during the data collection process for the verbal consent.
- Please discuss further on previous research and compare it to the study's findings in the Discussion section. Based on your comment, the section of Discussion has been revised and supplemented.
References
Stevens, J. (2002). Applied multivariate statistics for the social sciences. (4th ed.). Mahawah, NJ: Lawrence Erlbaum Associates.
Field, A. (2009). Discovering statistics using SPSS (3rd ed.). Thousand Oaks, CA: Sage Publications.
Wetzel, A. P. (2011). Factor analysis methods and validity evidence: a systematic review of instrument development across the continuum of medical education. Unpublished doctoral dissertation: Virginia Commonwealth University.
Yo, T., & Richardson, J. C. (2016). An exploratory factor analysis and reliability analysis of the student online learning readiness (SOLR) instrument. Journal of Asynchronous Learning Network, 19(5), 120-141.

Reviewer 3 Report
This cross-sectional survey of resident foreigners in South Korea examined the association between active, passive and social leisure activities and measures of self-efficacy and social adjustment. The study is felt to be very limited in scope and methodologically flawed as indicated by:
1. The title should reflect that the study was done in South Korea.
2. The report makes no mention of issues such as race and discrimination in the majority population towards minority resident foreigners. As a result, the research places the “blame” on resident foreigners for not experiencing “smooth social adjustment” to their new country of residence.
3. The sample was a convenience sample with no indication of the sampling frame and sampling selection process. As a result, the study findings cannot be generalized to the population of resident foreigners in South Korea.
4. The authors modify both their instruments measuring self-efficacy and social adjustment and only test the psychometric properties of the new instruments in the same sample that was used for primary research study. As a result, the reader has little confidence in the reliability and validity of modified instruments.
5. The authors fail to acknowledge that their cross-sectional study cannot explain the causal relationship between leisure activities and self-efficacy and social adjustment. Therefore, any recommendations about how to improve self-efficacy and social adjustment in foreign residents is premature.
Author Response
Reviewer 3.
- The title should reflect that the study was done in South Korea. Based on your comment, we have revised the title.
- The report makes no mention of issues such as race and discrimination in the majority population towards minority resident foreigners. As a result, the research places the “blame” on resident foreigners for not experiencing “smooth social adjustment” to their new country of residence. Based on your comment, we have added the issues you mentioned in the section of Introduction.
- The sample was a convenience sample with no indication of the sampling frame and sampling selection process. As a result, the study findings cannot be generalized to the population of resident foreigners in South Korea. We totally understand your concern. As we mentioned in the manuscript, this study applied the convenience sampling in order to collect research data from foreign residents in a racially homogeneous nation (i.e., South Korea). However, the convenience sampling technique has been implemented successfully in social science field, and this study investigating self-efficacy and social adjustment of foreign residents in the racially homogeneous nation could secure more in-depth information.
- The authors modify both their instruments measuring self-efficacy and social adjustment and only test the psychometric properties of the new instruments in the same sample that was used for primary research study. As a result, the reader has little confidence in the reliability and validity of modified instruments. We totally understand your comment. After minor modifications, a total of three professors in physical education, including one professor in sports psychology, were confirmed through the content validation process that there was no problem measuring self-efficiency and social adjustment of survey respondents who participated in the study. Additionally, as reported in the manuscript, all psychometric properties were also statistically satisfactory for this study.
- The authors fail to acknowledge that their cross-sectional study cannot explain the causal relationship between leisure activities and self-efficacy and social adjustment. Therefore, any recommendations about how to improve self-efficacy and social adjustment in foreign residents is premature. We understand your comment. This study was implemented based on comparative research design with a multivariate analysis of variance (MANOVA) to find differences in variables among groups (the research purpose). However, we could conduct future studies to find the causal relationship among significant factors for foreign residents.

Round 2
Reviewer 3 Report
I don't feel that the authors are able to address my concerns as the study design is seriously flawed
Author Response
Thank you for your comments.